# Functional Characterization of *PeVLN4* Involved in Regulating Pollen Tube Growth from Passion Fruit

**DOI:** 10.3390/ijms26052348

**Published:** 2025-03-06

**Authors:** Hanbing Yang, Xiuqing Wei, Lifeng Wang, Ping Zheng, Junzhang Li, Yutong Zou, Lulu Wang, Xinyuan Feng, Jiahui Xu, Yuan Qin, Yuhui Zhuang

**Affiliations:** 1College of Life Sciences, Haixia Institute of Science and Technology, Fujian Agriculture and Forestry University, Fuzhou 350002, China; 1220514011@fafu.edu.cn (H.Y.); 52305043020@fafu.edu.cn (L.W.); zhengping13@mails.ucas.ac.cn (P.Z.); lijunzhang@webmail.hzau.edu.cn (J.L.); 1220525052@fafu.edu.cn (Y.Z.); luluwanghn@163.com (L.W.); 15396098808@163.com (X.F.); 2Fruit Research Institute, Fujian Academy of Agricultural Sciences, Fuzhou 350013, China; weixiuqing47@foxmail.com (X.W.); xjhui577@163.com (J.X.)

**Keywords:** *Passiflora edulis*, *VLN* gene, actin filament, reproduction development

## Abstract

Passion fruit (*Passiflora edulis*), mainly distributed in tropical and subtropical regions, is popular for its unique flavor and health benefits. The actin cytoskeleton plays a crucial role in plant growth and development, and villin is a key regulator of actin dynamics. However, the mechanism underlying the actin filament regulation of reproductive development in passion fruit remains poorly understood. Here, we characterized a villin isovariant in passion fruit, *Passiflora edulis VLN4* (*PeVLN4*), highly and preferentially expressed in pollen. Subcellular localization analysis showed that PeVLN4 decorated distinct filamentous structures in pollen tubes. We next introduced *PeVLN4* into *Arabidopsis villin* mutants to explore its functions on the growing pollen tubes. *PeVLN4* rescued defects in the elongation of *villin* mutant pollen tubes. Pollen tubes expressing *PeVLN4* were revealed to be less sensitive to latrunculin B, and *PeVLN4* partially rescued defects in the actin filament organization of *villin* mutant pollen tubes. Additionally, biochemical assays revealed that PeVLN4 bundles actin filaments in vitro. Thus, PeVLN4 is an important regulator of F-actin stability and is required for normal pollen tube growth in passion fruit. This study provides a new insight into the function of the actin regulator villin involved in the reproduction development of passion fruit.

## 1. Introduction

The genus *Passiflora* is the largest in the family *Passifloraceae*, comprising around 500 species. Of these, *Passiflora edulis* (passion fruit) is a very popular species because of its important economic value, being widely planted in tropical and subtropical areas of the world [1]. The purple-fruited type *P. edulis Sims* and the yellow-fruited *P. edulis f. flavicarpa O. Deg.* are the two main common varieties of passion fruit with considerable economic value [1,2]. Passion fruit has essential eating quality due to its egg-shaped fruits with juiciness, nutritional values, and health benefits. Additionally, it has ornamental value due to its attractive flower [3,4,5]. Since the reproductive organs of passion fruit have great value, it is of great significance to explore the molecular mechanism involved in reproductive development of passion fruit.

Pollen tube growth is an important event in plant reproductive development. Pollen tubes grow rapidly in the form of tip growth [6]. During this rapid growth, pollen tubes are able to timely deliver sperm cells to the embryo sac to accomplish double fertilization in flowering plants [6]. The actin cytoskeleton is involved in numerous significant cellular processes, such as cell division and cell expansion like tip growth [7,8,9,10]. The actin cytoskeleton exhibits distinct spatial structures in the growing pollen tube [7,11]. The actin cytoskeleton has a specific spatial distribution in the growing pollen tube to perform specific functions, which is less abundant but highly dynamic at the tip region, and exists as longitudinal actin cables in the shank region of the pollen tube [7,12,13,14,15]. Multiple actin-based functions are related with the spatial arrangement and dynamics of actin structures, including actin nucleating, capping, depolymerizing, severing, and bundling [16,17,18,19]. Actin dynamics are modulated by different actin-binding proteins [13,20,21,22,23,24,25]. Among these, regulators like actin depolymerization factors (ADFs) are responsible for depolymerizing and severing single actin filaments [26,27,28]. Formin is one kind of actin nucleation factor responsible for the generation of linear actin bundles [29,30]. It contains formin homology domains responsible for nucleating the actin assembly from actin or actin–profilin complexes [29,31]. Several classes of ABPs bundle and cross-link F-actin to form and maintain higher-order actin filament structures, such as villins, LIMs, and fimbrins, as reported previously [12,32,33]. Dissecting how the actin cytoskeleton is modulated by various actin regulators to perform cellular functions is significant to understanding the cellular process of pollen tube growth.

Villin, a member of the villin/gelsolin/fragmin superfamily, is a key regulator of actin dynamics [34,35,36]. The regulatory functions of villin are related to its structural characteristics. It contains several gelsolin homology domains at the N-terminus and a headpiece domain at the C-terminus [34,37]. In previous studies, the first functionally characterized villin proteins were identified in *Lilium brownie*, namely 115-ABP and 135-ABP [38,39]. 135-ABP plays an important role in the regulation of F-actin in pollen tubes and F-actin organization in root hairs [38,39]. In *Arabidopsis*, there are five villin members involved in plant growth and development through the regulation of actin dynamics [34,40]. *AtVLN1* and *AtVLN4* have been identified to be associated with root hair length. The loss of function of *VLN1* or *VLN4*, respectively, leads to longer or shorter root hair [40,41]. *AtVLN2* and *AtVLN3* have been identified to be essential for normal plant organ morphogenesis. *vln2 vln3* double mutants exhibit curly organs and cannot grow with an erect habit [42,43,44]. *AtVLN5*, specifically expressed in pollen, has been reported to be essential in pollen tube growth [12]. In rice, OsVLN2 plays a key role in regulating plant architecture by modulating actin dynamics, as previously reported [45]. In cotton genome-wide analysis, *VLN* genes were identified [38], and GhVLN2 regulates the defense pathway by modulating actin cytoskeleton remodeling [46]. However, to date, there has been limited investigation of the actin cytoskeleton modulated by actin regulators in passion fruit.

In this study, we first functionally characterized the actin regulator villin in passion fruit. We conducted functional verification in vivo and biochemical assays in vitro to explore the function of *PeVLN4* in regulating actin filaments in growing pollen tubes. We propose that PeVLN4 plays a key role in pollen tube growth by modulating F-actin organization and stability in passion fruit. This study provides an insight into the function of an actin regulator villin participating in the reproduction development of passion fruit, and the theory supports molecular breeding improvement in passion fruit.

## 2. Results

### 2.1. Molecular Composition and Gene Expression of VLN Genes in Passion Fruit

In purple passion fruit, three villin members were identified in the genome data [47]. The three villin members contained conserved gelsolin homology domains. Two members contained six gelsolin homology domains from the N-terminus, and a headpiece domain (VHP) from the C-terminus. The other member contained fewer than six gelsolin homology domains, a VHP domain, and an extra transmembrane domain, implying it possibly functions at the cell membrane (Figure 1A). The genomic sequences were near 10 kb in length for these *VLN* genes of the purple passion fruit and contained several long introns (Figure 1B). The phylogenetic tree showed that the five *A. thaliana* villins (AtVLN1/2/3/4/5) were divided into three groups, and the three villins from passion fruit were divided into the same group of AtVLN4 and AtVLN5, implying that the three villins from passion fruit are more closely related to AtVLN4 and AtVLN5 (Figure 1C).

In terms of the gene expression of the *VLN* genes in passion fruit from the transcriptome data, *PeVLN4-1* was widely expressed in various nutritive and reproductive tissues but less expressed in stamens, while *PeVLN4* was the most highly expressed in the latter (Figure 1D). To further explore the gene expression in pollen, we collected pollen from purple passion fruit and performed fluorescent quantitative PCR to verify the gene expression. The data confirmed that *PeVLN4* was expressed the most in pollen compared to the other members (Figure 1E). Based on the expression data, we further explored the role of passion fruit *PeVLN4* in pollen tube growth.

### 2.2. Subcellular Localization Analysis of PeVLN4 in Pollen Tubes

In order to investigate the role of *PeVLN4* in pollen tubes, it is important to first detect where it is localized in the pollen tube. A construct containing green fluorescent protein (GFP) fused with the *PeVLN4* coding sequence (*PeVLN4-GFP*) was generated and then transiently expressed in the leaves of *Nicotiana benthamian* to examine the fluorescent signal. The image showed clear filamentous structures in the cytoplasm (Appendix A). Further, the construct was introduced into wild-type *Arabidopsis thaliana* (Col-0). As shown in Figure 2A, two transgenic lines were selected for further analysis. As shown in Figure 2B, Z-slices and integration analysis exhibited that PeVLN4 decorated filamentous structures in the shank as well as the apex of the pollen tube, which was similar to the subcellular localization of AtVLN5 reported in a previous study [48].

### 2.3. PeVLN4 Is Required for Normal Pollen Tube Growth

In previous studies, *AtVLN5* was expressed preferentially in pollen, and the *Arabidopsis vln5* mutant had defects in pollen tube elongation [12]. *AtVLN4* is lowly expressed in pollen (based on expression data from the TAIR website), and it mainly functions in nutrient tissues [41]. *PeVLN4, AtVLN4*, and *AtVLN5* genes are evolutionarily more closely related compared to *AtVLN2* and *AtVLN3* (Figure 1C). In order to investigate in vivo the function of *PeVLN4* in pollen tube growth, we next introduced *PeVLN4* into the *Arabidopsis vln5* mutant to determine whether it could rescue defects in the pollen tube elongation of the *vln5* mutant. The construct of *pK7FWG2M-PeVLN4pro:PeVLN4-GFP* was introduced into the *vln5* mutant for further research. We conducted PCR verification of *PeVLN4* in transgenic lines expressing *PeVLN4* (Figure 3A). To assess the extent to which *PeVLN4* can restore actin defects in *vln5* mutant pollen tubes, we performed fluorescent quantitative PCR to select two transgenic lines expressing *PeVLN4* at levels comparable to native *VLN5* transcripts in wild-type Col-0 for further experiments (Figure 3B). We subsequently tracked individual growing pollen tubes using light microscopy. Our findings revealed that wild-type pollen tubes were noticeably longer than *vln5* mutant pollen tubes but almost the same as the transgenic line pollen tubes harboring *PeVLN4* after 3 h of pollen germination on a standard medium (Figure 3C). We subsequently observed pollen tube growth after 40 min and collected images (Figure 3D).

To quantify the growth rate of pollen tubes, the statistical result of pollen tube length showed that the average length of *vln5* pollen tubes was significantly shorter than those of the wild-type (*p* < 0.01); however, the average length of the transgenic line pollen tubes was not significantly different from the wild-type (*p* > 0.05). The average growth rate (±SD) of the wild-type Col-0, *vln5*, and the transgenic line pollen tubes was 223.1 ± 24.26 μm/h, 127.7 ± 11.44 μm/h, and 199.2 ± 2.46 μm/h 200.7 ± 6.73 (*n* = 3), respectively (Figure 3E). These data illustrate that *PeVLN4* is required for normal pollen tube growth.

### 2.4. Pollen Tubes Expressing PeVLN4 Are Less Sensitive to LatB Treatment

We attempted to visualize whether F-actin in *PeVLN4* transgenic line pollen tubes responded to the actin drug Latrunculin B (LatB) differentially compared to the *vln5* mutant. Latrunculin B prevents actin polymerization by binding to monomeric actin with high affinity [12]. Thus, we applied it for the detection of dynamic actin filament arrays. Actin filaments were treated with 100 nM LatB for 30 min and then stained with Alexa-488 phalloidin. As shown in Figure 4A, actin filaments became noticeably shorter and less abundant in the wild-type Col-0 pollen tube, and F-actin fragments appeared smaller and darker in *vln5* mutant pollen tubes after the LatB treatment, as reported in a previous study on *Arabidopsis* [12]. Compared to the *vln5* mutant, F-actin fragments appeared brighter in the *vln5* mutant harboring *PeVLN4* pollen tubes. The average pixel intensity for pollen tubes from fluorescence images was used to assess the relative actin amount.

The amount of F-actin in the untreated wild-type Col-0, *vln5* mutant, and *PeVLN4* transgenic line pollen tubes was set to 100%. The relative amount of F-actin was reduced to 47.6%, 70.4%, 59.3%, and 58.1% of that in the treated wild-type Col-0, *vln5* mutant, and two transgenic line pollen tubes expressing *PeVLN4*, respectively (Figure 4B). This demonstrates that transgenic line pollen tubes expressing *PeVLN4* have better resistance to the LatB treatment than *vln5* mutant pollen tubes (*p* < 0.01). These results indicate that PeVLN4 is required for stabilizing actin filaments in pollen tubes.

### 2.5. PeVLN4 Is Required for the Organization of Actin Filaments in Pollen Tubes

In previous studies, *AtVLN2* and *AtVLN5* were expressed relatively highly in pollen compared to other villin members (based on expression data from the TAIR website), and the *vln2 vln5* double mutant revealed obvious defects in F-actin in pollen tubes [48], while the *vln5* mutant exhibited no obvious defects in F-actin distribution [12]. In order to explore the function of PeVLN4 in the organization of the actin cytoskeleton in pollen tubes, we subsequently introduced *PeVLN4* into the *vln2 vln5* double mutant expressing *Lat52:Lifeact-GFP* to observe the extent to which *PeVLN4* can restore actin defects in *vln2 vln5* pollen tubes. We performed fluorescent quantitative PCR to choose two transgenic lines expressing *PeVLN4* at levels comparable to native *AtVLN5* transcripts in the wild-type for the following experiments (Appendix A). As shown in Appendix A, *PeVLN4* partially restored the growth rate (Appendix A), and in addition, partially restored the defects in the width of *vln2 vln5* pollen tubes (Appendix A). The visualization of F-actin stained with Alexa-488-phalloidin revealed well-organized structures in wild-type Col-0 pollen tubes, and in contrast, actin filaments appeared disordered and very thin in *vln2 vln5* pollen tubes. However, actin filaments appeared relatively better organized in *vln2 vln5* pollen tubes harboring *PeVLN4* (Figure 5A).

To quantify the differences among the genotypes, the thickness of actin cables in the shank was measured and found to be considerably reduced in the *vln2 vln5* mutant, but the induction of *PeVLN4* partially restored the defect in *vln2 vln5* pollen tubes (Figure 5B). In addition, the angle of actin cables to the growth axis of *vln2 vln5* pollen tubes increased markedly. However, the introduction of *PeVLN4* in *vln2 vln5* partially restored the actin defects in *vln2 vln5* pollen tubes, illustrating its significant role in maintaining an orderly distribution of the actin cytoskeleton in pollen tubes (Figure 5C).

### 2.6. PeVLN4 Bundles Actin Filaments In Vitro

Since PeVLN4 appeared to maintain the stability of actin filaments in pollen tubes and be required for normal pollen tube growth, we explored the PeVLN4 regulation of actin filament dynamics in vitro. Recombinant *PeVLN4* was expressed in *E. coli* and purified using the adsorption effect of Ni-beads and His tags (Figure 6A). Most villins in previous studies exhibited actin filament bundling activity, thus it is reasonable to speculate that PeVLN4 will bundle actin filaments in vitro. To test it, we employed a low-speed co-sedimentation assay. As shown in Figure 6B,C, in lane 2, actin alone did not sediment noticeably under these conditions, but the presence of PeVLN4 increased the amount of sedimentable actin. PeVLN4 bundled actin filaments in a dose-dependent manner (Figure 6B,C).

To confirm the actin bundling activity of PeVLN4, actin filaments stained with rhodamine-phalloidin in the presence and absence of PeVLN4 were visualized directly under fluorescence light microscopy. Actin filaments appeared to be individual structures in the absence of PeVLN4; however, in the presence of 1 μM PeVLN4, actin filaments were organized into massive bundles (Figure 6D). These results demonstrate that PeVLN4 bundles actin filaments to form higher structures.

## 3. Discussion

Passion fruit is an important tropical fruit with great economic value. To explore the related molecular mechanisms involved in reproduction development is important for understanding the reproduction of passion fruit. In the process of fertilization in flowering plants, the pollen tube grows through a long distance in the style, and sperm cells were delivered to the ovule to fulfill double fertilization [7]. Pollen tube growth depends on actin dynamics and the distinct F-actin spatial distribution modulating by various actin regulators [7,20]. Investigating actin regulators in growing pollen tubes for passion fruit makes great sense for understanding the plant reproductive development of passion fruit.

In this study, we functionally characterized the actin regulator villin in passion fruit. There are three villin homology members containing conserved gelsolin domains and a headpiece domain in passion fruit. *VLN* genes in passion fruit contains many long introns, around 10 kb in length (Figure 1B); however, *Arabidopsis VLN* genes contain less introns and around 5 kb in length [34]. Previous studies have found that the presence of introns can promote the expression of the mRNA containing them, and some of the introns encode snoRNA and miRNA [49,50]. The *VLN* genes of passion fruit contain so many introns, and their potential special functions remain unknown. It is worth further investigating these introns’ regulatory effects on the function of *VLN* genes. Additionally, the genomic sequences of *PeVLN4* and *PeVLN4-2* are very similar (Figure 1B), and their expression profiles from transcriptome data are also similar (Figure 1D). However, qRT verification showed that *PeVLN4-2* was at a very low level in pollen, in contrast with *PeVLN4* (Figure 1B,E); the reason for which requires further investigation. Among these *VLN* genes in passion fruit, *PeVLN4* was identified to be relatively highly expressed in pollen (Figure 1). We subsequently introduced *PeVLN4* into the model plant species *Arabidopsis* and found that PeVLN4 decorated obvious filament structures in the cytoplasm of the pollen tube (Figure 2), just as the subcellular localization of *Arabidopsis* VLN5 in the pollen tube [48]. It implied that PeVLN4 played an important role in the formation of actin cables in the shank of pollen tubes, as the key role of AtVLN5 in the *Arabidopsis* pollen tube. Biochemical assays in vitro directly verified that PeVLN4 bundled F-actin to form higher-ordered structures (Figure 6). It implies that passion fruit PeVLN4 is essential for the formation of long actin cables in the growing pollen tube, just as AtVLN5 is crucial in the *Arabidopsis* pollen tube [12,48,51].

Functional identification in vivo revealed that *PeVLN4* rescued defects in the pollen tube growth of the *vln5* mutant (Figure 3) and partially rescued the defects in maintaining the stability of actin filaments in pollen tubes (Figure 4). The visualization of actin filaments in pollen tubes revealed that *PeVLN4* partially rescued the defects in the F-actin organization of the *vln2 vln5* mutant (Figure 5). These data imply that passion fruit PeVLN4 is a key regulator in F-actin organization in the growing pollen tube. In previous studies, *AtVLN5* fully rescued the defects of the *vln2 vln5* pollen tube phenotype [48]. Although *PeVLN4* just partially rescued the defects of the *vln2 vln5* mutant, it also illustrates that passion fruit PeVLN4 retains a certain functional conservatism in the regulation of the actin dynamics in growing pollen tubes as AtVLN5 functions. These results provide an inspiration to dissect the villin members in different species.

In our study, we confirm that PeVLN4 is crucial for pollen tube growth by modulating actin organization. It suggest that PeVLN4 is an important regulator during the reproductive development of passion fruit. There are many scientific problems in the reproductive development of passion fruit to be explored. Purple passion fruit has self-compatibility, while yellow passion fruit has self-incompatibility, which requires exogenous pollination to set fruit [52]. In previous studies, in self-incompatible pollen, actin depolymerization was sufficient to induce programmed cell death [53]. It has been reported that villins play a key role in regulating F-actin depolymerization during mimicking the self-incompatibility response [54]. Thus, based on our study, it is significant to further explore *VLN* genes’ roles in regulating self-incompatibility in passion fruit, which is helpful to understand the mechanism of self-incompatibility in passion fruit. Furthermore, passion fruit is easy to fall at high temperatures, and its growth requires strict climatic factors. At high temperature, the fruit setting rate is significantly reduced, and high temperatures affect the development of young buds; buds will fall in severe cases or neither flower nor fruit. Low temperatures affect flowering, pollination, and fertilization in passion fruit [55,56]. Based on the characterization of *PeVLN4* in the reproductive development of passion fruit, it is worth exploring whether *VLN* genes are related to those processes, which will provide an insight into the mechanisms of reproductive development in passion fruit that are affected by temperature.

In conclusion, our findings elucidate the function of passion fruit *PeVLN4* in growing pollen tubes. PeVLN4 is the crucial regulator for modulating the actin filaments in pollen tubes for normal pollen tube growth in passion fruit. This study first provides an insight into the function of an actin regulator in modulating the actin cytoskeleton during the plant reproductive development of passion fruit and offers theory for supporting molecular breeding improvement in passion fruit.

## 4. Materials and Methods

### 4.1. Identification, Sequence Analysis, and Expression Analysis of VLNs from Passion Fruit

The genomic sequences of *PeVLNs* were extracted from the genome data reported previously [47]. The exons and introns were indicated in https://gsds.gao-lab.org/ (accessed on 21 July 2024). Full-length amino acid sequences were submitted to SMART (https://smart.embl.de/) (accessed on 20 October 2022) to analyze the protein structures. The amino acid sequences of PeVLNs and AtVLNs [32] were aligned via ClustalW followed by the neighbor-joining method using MEGA version 6 to generate the phylogenetic tree. The transcriptome data were derived from previous work [57], which was deposited in the China National GeneBank Database (CNGBdb) under the accession number CNP0006659 (various vegetative tissues and fruit development data) and CNP0005768 (various floral organ development data). TPM values were applied to TBtools (version 1.120) to generate heatmaps.

### 4.2. Construction and Plant Materials

To generate *PeVLN4* complementary lines, the native *PeVLN4* promoter that was 2000 bp in length and cDNA were amplified and cloned into the vector *pK7FWG2M-GFP*. Desired *pK7FWG2M-PeVLN4pro:PeVLN4-GFP* and *pK7FWG2M-PeVLN4pro:PeVLN4* plasmids were generated, and the resulting constructs were introduced into *Arabidopsis* plants Col-0, *vln5* (*vln5-1*) [12], and *vln2 vln5* double mutant (expressing *Lat52pro:Lifeact-GFP*) [48] by *Agrobacterium*-mediated transformation to acquire *PeVLN4pro:PeVLN4-GFP;WT*, *PeVLN4pro:PeVLN4-GFP;vln5*, and *PeVLN4pro:PeVLN4;vln2 vln5* (*Lat52pro:Lifeact-GFP*) transgenic lines. To generate the transient expression construct, *PeVLN4* cDNA was amplified and cloned into the vector *pCambia1301-35Spro*, and *pCambia1301-35Spro-PeVLN4-GFP* was generated. All primer pairs are listed in Appendix A. The purple passion fruit variety was used for this study.

### 4.3. Quantitative Real-Time PCR

Total RNA was extracted from the pollen of purple passion fruit and *Arabidopsis* by an RNA extraction kit (HiPure Plant RNA Mini Kit, Magen, Guangzhou, China). The transcripts of three *VLN* genes were amplified from the passion fruit with the primer pairs in Appendix A; the transcript of *PeVLN4* was amplified from *Arabidopsis* transgenic lines with the primer pairs in Appendix A using SYBR Green I (Vazyme, Nanjing, China) by QuantStudio^TM^ 1 Plus Real-Time fluorescence quantification PCR System (Applied Biosystems, Waltham, MA, USA). *EF1A* (*elongation factor 1 alpha*) was used as the internal control in passion fruit, with the primer pairs in Appendix A; *eIF4A* was used as the internal control in *Arabidopsis* (Appendix A). The relative expression level of *VLN* genes was calculated by the method of 2^ΔCt^ [51], ΔCt = Ct (*eIF4A* or *EF1A*) − Ct (*VLN*).

### 4.4. Pollen Tube Growth and Quantification

Pollen was harvested from each genotype’s flowers and cultivated on pollen germination medium (1 mM Ca(NO_3_)_2_, 1 mM CaCl_2_, 1 mM MgSO_4_, 0.01% (*w*/*v*) H_3_BO_3_, 18% (*w*/*v*) sucrose, and 0.8% (*w*/*v*) agar, pH 6.9–7.0 at 28 °C, as the previous method [48]. After culturing for 3 h, images of pollen tubes were captured under a light microscope (NEXCOPE, Ningbo, China) equipped with a 4× (0.13 NA) or 10× (0.3 NA) objective. The images were acquired using Nscope 2.1 software (NEXCOPE, Ningbo, China). To quantify the pollen tube growth rate, more than 30 pollen tubes were measured for each genotype using Image J software (version 1.4.3.67).

### 4.5. LatB Treatment and Staining of Actin Filaments in Pollen Tubes

To observe the actin filaments in pollen tubes, pollen harvested from each genotype’s flowers was cultured on a pollen germination medium at 28 °C. After pollen tubes reached around 100–200 μm, we first treated pollen tubes on the medium with 100 nM LatB for 30 min, and subsequently treated pollen tubes on the medium with 300 μM N-(maleimidobenzoyloxy)-succinimide (MBS) at 28 °C for 1 h, then 150 mM MBS containing 0.05% NP-40 for 10 min at room temperature, and after that, pollen tubes were washed with TBSS containing 0.05% NP-40 (50 mM Tris, 200 mM NaCl, pH 7.4) twice, 10 min each time at room temperature, and washed a last time with TBSS without NP-40 for 10 min. F-actin was stained with 132 nM Alexa-488 phalloidin in the TBSS without NP-40 overnight at 4 °C. Images were captured by a confocal microscope (Leica Microsystems CMS GmbH, Wetzlar, Germany) equipped with a 100× oil objective (1.4 numerical aperture). The fluorescent phalloidin was excited using a 488 nm laser. The z-series images were collected with LAS X software (version 3.5.7.23225, Leica, Wetzlar, Germany) with the step size set at 0.5 μm. Images were analyzed by Image J software (version 1.4.3.67). The angle and width of F-actin in the shank of pollen tubes (15–30 μm from the tip) were measured and quantified by Image J software. At least 200 actin filaments from more than ten pollen tubes were measured to quantify the angles to the growth axis of pollen tubes, as described previously [58]. At least 100 actin filaments from more than ten pollen tubes were measured to quantify the widths.

### 4.6. Visualization of Intracellular Localization of PeVLN4 in Pollen Tubes

Pollen harvested from flowers was cultured on a solid pollen germination medium. After pollen tubes reached around 100–200 μm, they were observed under a confocal microscope (Leica Microsystems CMS GmbH) equipped with a 100× oil objective (NA, 1.4). GFP was excited by a 488 nm laser, and the images were collected by LAS X software with a step size of 0.5 μm. Images were analyzed by Image J (version 1.4.3.67).

### 4.7. Transient Expression Assay

The construct of *pCambia1301-35Spro:PeVLN4-GFP* was first introduced into *Agrobacterium tumefaciens* GV3101 and then infiltrated into the expanded leaves of *N. benthamiana*, as the previous method [59]. After incubation in the growth room for 96 h, the injected leaves were collected, and the images were collected by LAS X software.

### 4.8. Protein Production

The plasmid of *pET23b-PeVLN4* was transformed into *E. coli* BL21 (DE3) strains, induced with 0.4 mM isopropyl β-D-thiogalactoside (IPTG) for 12–16 h at 16 °C, and then broken by sonication in a 5 mM imidazole solution (25 mM Tris-HCl, pH 8.0, 250 mM KCl, 5 mM imidazole) containing 1 mM PMSF and a little lysozyme. The sample was centrifuged at 4 °C under 20,442× *g* for 40 min, and the collected supernatant was loaded onto a column containing Ni-TED Sepharose 6FF (His-Tag) (Sangon Biotech, Shanghai, China, C610030-0025). The sample was subsequently washed with the 5 mM imidazole solution above, and the bound protein was eluted with buffer (25 mM Tris-HCl, pH 8.0, 250 mM KCl, 600 mM imidazole). The eluted proteins were dialyzed in 5 mM Tris-HCl (pH 8.0) and finally stored in −80 °C. Actin was derived from rabbit muscle, as previously reported [59].

### 4.9. Low-Speed F-Actin Co-Sedimentation Assay

A low-speed F-actin co-sedimentation assay was performed as previously reported [60]. In brief, various concentrations of PeVLN4 were incubated with 3 μM F-actin for 30 min at room temperature in a 1xKMEI solution containing 50 mM KCl, 1 mM MgCl_2_, 1 mM EGTA, and 10 mM imidazole, pH 8.0. The samples were subjected to centrifugation under 13,600× *g* for 30 min at 4 °C. The supernatant and pellet of the samples were respectively separated by 10% SDS-PAGE, and stained with Coomassie Brilliant Blue R. The amount of actin in the supernatant and pellet was quantified by Image J software (version 1.4.3.67).

### 4.10. Fluorescence Light Microscopy of Actin Filaments

Actin filaments stained with rhodamine-phalloidin were directly observed by fluorescence light microscopy as reported previously [59]. In brief, PeVLN4 was incubated with 4 μM F-actin in 1xKMEI containing an equal molar amount of rhodamine-phalloidin for 30 min at room temperature. The mixtures were diluted in fluorescence buffer (50 mM KCl, 1 mM MgCl_2_, 10 mM imidazole, pH 7.0, 100 mM DTT, 100 µg/mL glucose oxidase, 20 µg/mL catalase, 15 mg/mL glucose, and 0.5% methylcellulose) and captured under a ZEISS fluorescence microscope (Oberkochen, Germany) equipped with a 63× (1.4 NA) oil objective. Images were acquired by ZEN 2 software (version 2.3, Carl Zeiss, Oberkochen, Germany). The final F-actin concentration was 1 μM in fluorescence buffer.

## Figures and Tables

**Figure 1 ijms-26-02348-f001:**
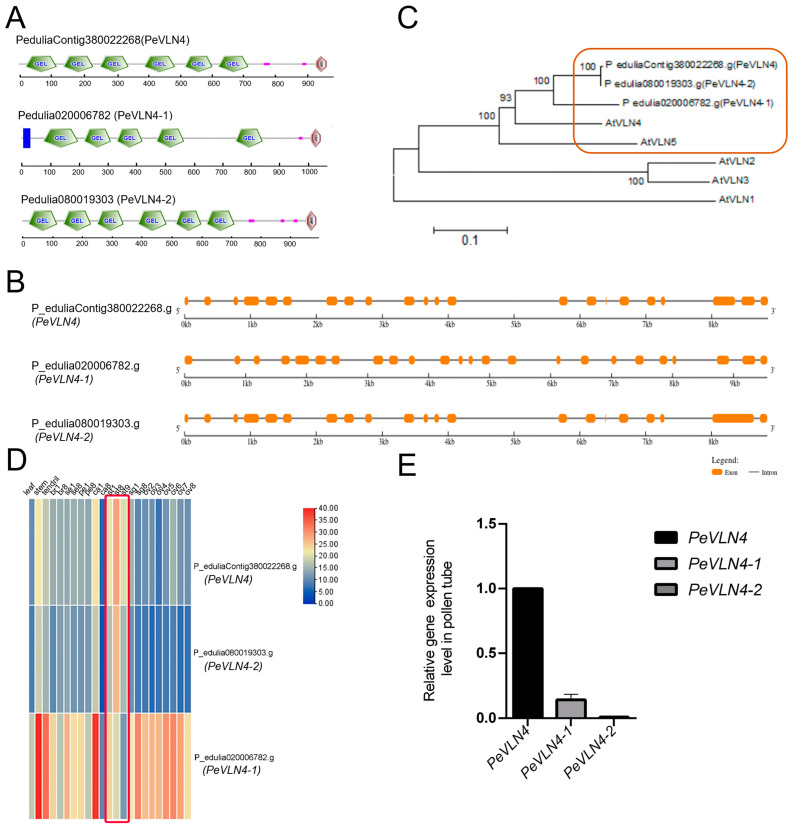
Molecular organization and gene expression analysis of villins in passion fruit. (**A**) The phylogenetic tree of villins in *Arabidopsis* and passion fruit. To generate the phylogenetic tree, amino acid sequences were aligned via ClustalW followed by the neighbor-joining method using MEGA version 6. (**B**) The protein domain organization of PeVLNs. Full-length amino acid sequences were analyzed using SMART (https://smart.embl.de/) (accessed on 20 October 2022). (**C**) The genomic sequence analysis of PeVLNs. The orange frame indicates the exon region, and the black line indicates the intron region, via https://gsds.gao-lab.org/ (accessed on 21 July 2024). (**D**) Transcriptome data analysis represented on a heatmap by TBtools (version 1.120). Various tissues at different developmental stages: leaf; stem; tendril; br, bract; se, sepal; pe, petal; ca, corona filament; st, stamen; sg, stigma; ov, ovule; the numbers indicate developmental stages. The numbers 1 and 2 represent the early stages, and 8 indicates a late stage. (**E**) qRT-PCR analysis was performed to determine the relative amount of *PeVLN* transcripts in the pollen from purple passion fruit. The expression level of the *EF1A* gene was used as an internal control. Data were presented as the mean ± SD (*n* = 3).

**Figure 2 ijms-26-02348-f002:**
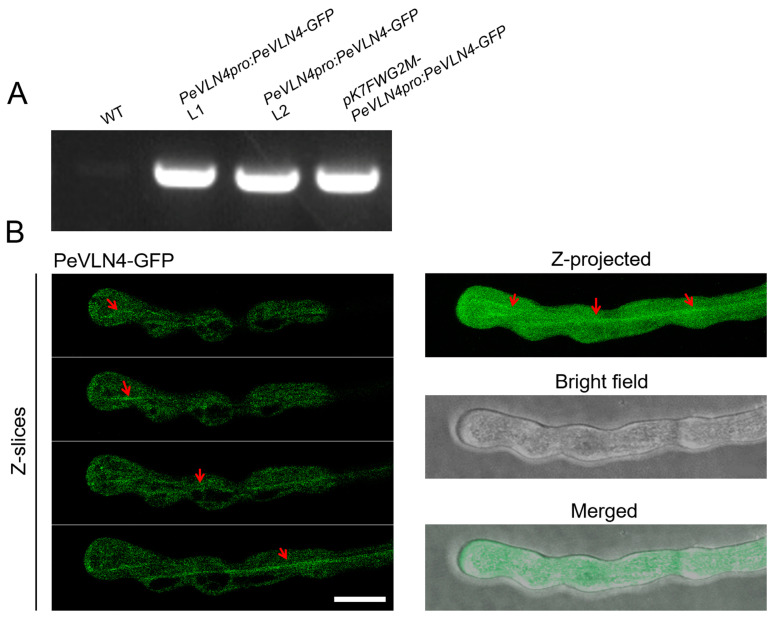
PeVLN4 decorates filamentous structures in a pollen tube. (**A**) PCR verification of *PeVLN4-GFP* in transgenic lines. (**B**) Visualization of the subcellular localization of PeVLN4-GFP in a pollen tube. The construct of *pCambia1301-PeVLN4pro:PeVLN4-GFP* was transformed into *Arabidopsis thaliana*. Filamentous structures in the shank or apex are indicated by red arrows. Bar = 5 μm.

**Figure 3 ijms-26-02348-f003:**
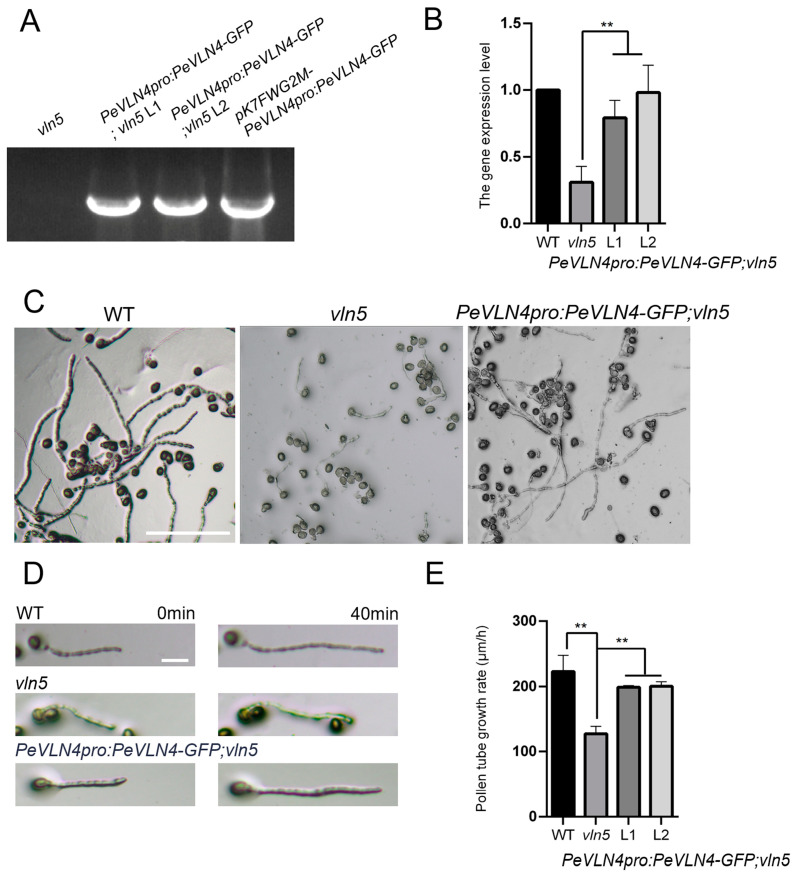
*PeVLN4* rescues defects in *villin* mutant pollen tubes. (**A**) PCR verification for wild-type Col-0, *vln5* mutant, transgenic lines expressing *PeVLN4*, and the plasmid as a positive control. Lanes 1, 2, 3, and 4 represent *vln5*, transgenic lines 1 and 2, and the plasmid. (**B**) Quantitative real-time PCR analysis for Col-0, *vln5* mutant, and transgenic lines expressing *PeVLN4*. The expression level of the *eIF4A* gene was used as an internal control. Data are presented as the mean ± SD (*n* = 3); ** *p* < 0.01 (Student’s *t*-test). (**C**) Micrographs of pollen tubes after germination for 3 h. Pollen was isolated from plants with the following genotypes: wild-type Col-0, *vln5* mutant, and homozygous transgenic lines. Bar = 100 μm. (**D**) Micrographs of pollen tubes at two time points. Pollen was isolated from plants with the following genotypes: wild-type Col-0, *vln5* mutant, and homozygous transgenic lines. Bar =50 μm. (**E**) Quantification of the growth rate of the pollen tubes. Data were presented as the mean ± SD (*n* = 3). Pollen tube growth rate was calculated for Col-0, *vln5* mutant, and homozygous transgenic lines. ** *p* < 0.01 (Student’s *t*-test).

**Figure 4 ijms-26-02348-f004:**
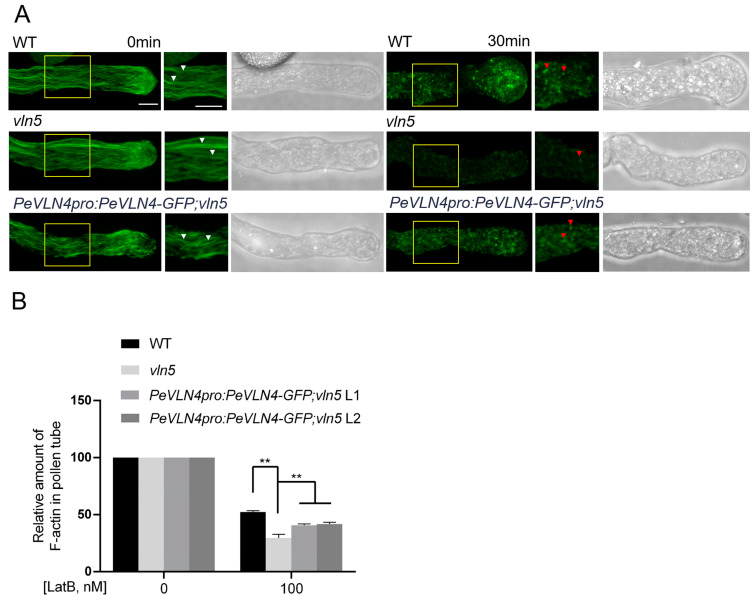
Visualization of F-actin in pollen tubes responding to Latrunculin B treatment. (**A**) Images of actin filaments stained with Alexa-488 phalloidin in pollen tubes. Both control and 100 nM LatB-treated pollen tubes with a bright field are presented. Bar = 5 μm. The rectangular box indicates the region for enlargement in the pollen tube. In the enlargement images, the white arrows indicate actin filaments in untreated pollen tubes at 0 min, and the red arrows indicate actin fragments in pollen tubes treated with LatB after 30 min. (**B**) Quantification of the relative amount of F-actin in pollen tubes. The amount of F-actin in untreated pollen tubes was normalized to 100. At least 10 pollen tubes were measured for each sample. Data represent the mean ± SE; ** *p* < 0.01 (Student’s *t*-test).

**Figure 5 ijms-26-02348-f005:**
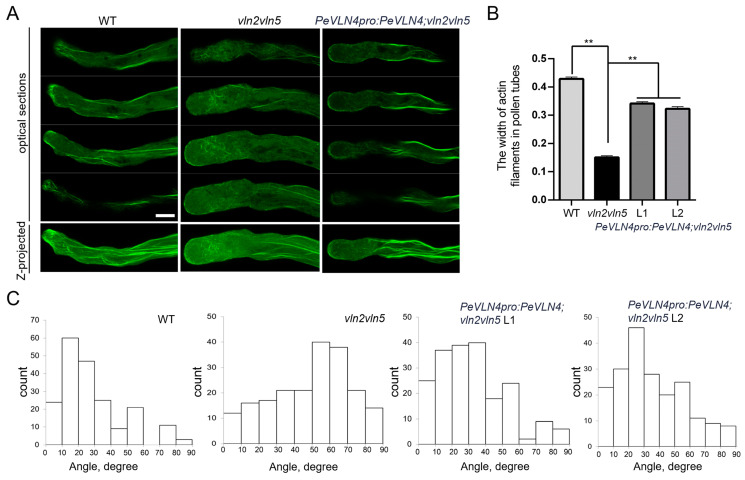
Organization of actin filaments in each genotype’s pollen tubes. (**A**) Images of actin filaments in pollen tubes. Both the projection images and optical sections of each genotype were presented; bar = 5 μm. (**B**) Quantification of actin cable width in the shank (15–30 μm) of pollen tubes. At least 100 actin cables were measured for each genotype. Data represent the mean ± SE; ** *p* < 0.01 (Student’s *t*-test). (**C**) Histogram of shank-localized (15–30 μm) F-actin angles to the growth axis of pollen tubes. About 200 actin cables were measured for each genotype.

**Figure 6 ijms-26-02348-f006:**
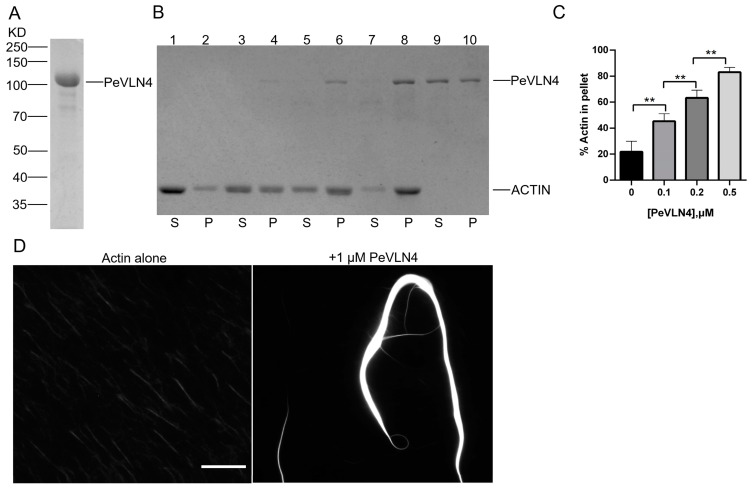
PeVLN4 bundles actin filaments in vitro. (**A**) SDS-PAGE analysis of purified PeVLN4 extracted from *E.coli*. and stained with Coomassie blue. (**B**) An F-actin low-speed co-sedimentation assay was employed to detect the bundling activity of PeVLN4. Samples for the supernatant (S) and pellet (P) were separated by SDS-PAGE; lanes 1, 3, 5, 7, and 9 represent the supernatants of actin alone, actin plus 100 nM PeVLN4, actin plus 200 nM PeVLN4, actin plus 500 nM PeVLN4, and PeVLN4 alone, respectively. Samples in lanes 2, 4, 6, 8, and 10 represent the pellets of actin alone, actin plus 100 nM PeVLN4, actin plus 200 nM PeVLN4, actin plus 500 nM PeVLN4, and PeVLN4 alone, respectively. (**C**) Quantification of the low-speed co-sedimentation assay. Data are presented as the mean ± SD (*n* = 3); ** *p* < 0.01 (Student’s *t*-test). (**D**) A fluorescence microscopy assay was employed for the visualization of actin filaments: 1 μM PeVLN4, 1 μM F-actin. Bar = 10 μm.

## Data Availability

The data are available in the article, Appendix A, and online repositories. The genome data and RNA-seq data in this study were deposited in the National Genome Data Center (NGDC) (https://ngdc.cncb.ac.cn/) (accessed on 17 October 2022) under the accession number GWHAZTM00000000 and the China National GeneBank Database (CNGBdb) under the accession numbers CNP0006659 and CNP0005768.

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
