# Peer review of "Functional Characterization of PeVLN4 Involved in Regulating Pollen Tube Growth from Passion Fruit"

_ijms, 2025, doi:10.3390/ijms26052348_

Round 1
Reviewer 1 Report
Comments and Suggestions for Authors
Dear authors and the editor,
This is a novel study of PeVLN4 on passion fruit, which is good for understanding the pollen tubes growth and breeders of passion fruit. But there are still some problems to be improved.
Q1: Please accurately definite the expressions of gene and protein of VLN4 in passion fruit with the italic and upright capital letters.
Q2: Just based on sequence analysis of VLNs from Passion Fruit, not forward genetics analysis based on populations, which seems to be a lack of direct scientific evidence, although there is the identification of PeVLN4 in Arabidopsis.
Q3: Fig 4A, should be improved for better distinguish the images of 210 actin filaments stained with Alexa-488 phalloidin in pollen tubes, at 0 min and 30 min.
Q4: Line 211, there is lack of spacing between (B) and Quantification, please check the whole MS.
Q5: The genetic transformation plants seem to lack three replicates and statistical analysis in Figure 1, 3 and 4, please check the data.
Q6: Please summarize the key findings and innovations, as well as the important conclusions in abstract.
So in my opinion, this paper should be reconsidered after minor revisions.
Comments on the Quality of English LanguageThat's Fine.
Reviewer 2 Report
Comments and Suggestions for Authors
Functional characterization of PeVLN4 involved in regulating pollen tube growth from passion fruit
Yang et al
Title
The title represents well the content of the manuscript but I do not understand the name of the gene PeVLN4
Abstract
Line 16: However, it remains poorly understood that the mechanism… DELETE “that”
Keywords
I suggest using keywords different from the words present in the title to improve the web engine topic research
Introduction
Lines 47-48: It reveals less abundant but highly dynamic at the tip of the pollen tube, and it is presented as longitudinal actin cables in the shank of the pollen tube. I understand that the authors mean that there is a spatial distribution but the sentence is not clear
Lines 77-78: In cotton, genome-wide analysis of VLN genes were identified [40]… Rewrite: In cotton genome wide analysis, VLN genes were identified
Line 84: PeVLN4. Is this the name of the gene the authors propose for the passion fruit villin expressed in the pollen tube?
Results
Line 91: In purple passion fruit, the founding three villin members were extracted…Rewrite: In purple passion fruit, the founding three villin members were extracted
Line 154: to determine whether it could rescue defects in vln5 Unclear sentence
Discussion
Lines 280-282: Redundancy. Already present in the introduction
Material and methods
Lines 358-359: The transcriptome data was derived from previous work… Add reference of the previous work
Line 364: in paragraph 4.2 information on the passion fruit genotype should be included because passion fruit is missing in the plant material!
Line 378: Total RNA was extracted from pollen of purple passion fruit or Arabidopsis…Replace with: Total RNA was extracted from the pollen of purple passion fruit and Arabidopsis
Line 379: add the extraction Kit name
Line 383: EF1A should be an acronym of what
Line 389: I don't understand what a genotype’s “flower” means. Did the authors use more than one genotype for their analysis?
Supplementary material
Line 365: The captions of the supplementary materials are insufficient to allow the reader to understand what they represent. A Captions file should be included in supplementary Material.
